# Antioxidant and Anti-Inflammatory Effects of Traditional Medicinal Plants for Urolithiasis: A Scoping Review

**DOI:** 10.3390/plants14132032

**Published:** 2025-07-02

**Authors:** Brenda Pacheco-Hernández, Teresa Ayora-Talavera, Julia Cano-Sosa, Lilia G. Noriega, Neith Aracely Pacheco-López, Juan M. Vargas-Morales, Isabel Medina-Vera, Martha Guevara-Cruz, Rodolfo Chim-Aké, Ana Ligia Gutiérrez-Solis, Roberto Lugo, Azalia Avila-Nava

**Affiliations:** 1Hospital Regional de Alta Especialidad de la Península de Yucatán, Servicios de Salud del Instituto Mexicano del Seguro Social para el Bienestar (IMSS-BIENESTAR), Mérida 97130, Yucatán, Mexico; brenda.pachecohernandez@gmail.com (B.P.-H.); rodolfochim@hotmail.com (R.C.-A.); ganaligia@gmail.com (A.L.G.-S.); roberto.lugo.gomez@gmail.com (R.L.); 2Centro De Investigación y Asistencia en Tecnología y Diseño del Estado de Jalisco, A. C. Subsede Sureste, Parque Científico Tecnológico de Yucatán, Sierra Papacal-Chuburná Puerto, Mérida 97302, Yucatán, Mexico; tayora@ciatej.mx (T.A.-T.); jcano@ciatej.mx (J.C.-S.); npacheco@ciatej.mx (N.A.P.-L.); 3Departamento de Fisiología de la Nutrición, Instituto Nacional de Ciencias Médicas y Nutrición Salvador Zubirán, Ciudad de México 14080, Mexico; lgnoriegal@gmail.com (L.G.N.); martha.guevarac@incmnsz.mx (M.G.-C.); 4Facultad de Ciencias Químicas, Universidad Autónoma de San Luis Potosí, San Luis Potosí 78300, Mexico; juan.vargas@uaslp.mx; 5Departamento Metodología de Investigación, Instituto Nacional de Pediatría, Ciudad de México 04530, Mexico; isabelj.medinav@gmail.com

**Keywords:** kidney stone, traditional medicine, bioactive compounds, oxidative stress

## Abstract

Urolithiasis (UL) is the presence of stones in the kidneys or urinary tract; its prevalence has increased worldwide. Thus, strategies have been sought to reduce it and one of them is the use of medicinal plants due to their accessibility, low cost, and cultural traditions. Studies on traditional medicinal plants in UL mainly documented results of litholytic and urinary parameters. Although, stone formation is related to oxidative stress and inflammation, and only a few studies are focused on these types of biomarkers. Thus, the aim of the present review was to summarize studies showing the antioxidant and anti-inflammatory effects of traditional medicinal plants used in UL management. We performed a scoping review; the database sources used were MEDLINE/PubMed, Google Scholar, SpringerLink, Scielo and Redalyc. From a total of 184 studies screened, six were included from China (2), India (3), and Corea (1). These studies have shown the antioxidant and anti-inflammatory effects of traditional medicinal plants, including *Glechoma longituba* (*G. longituba*), *Bergenia ligulate* (*B. ligulate*), *Lygodium japonicum* (*L. japonicum*), *Citrus limon* (*C. limon*), *Xanthium strumarium* (*X. strumarium*) and *Tribulus terrestris* (*T. terrestris*). They have also described their molecular mechanism of antioxidant and anti-inflammatory effects through the activation of antioxidant genes induced by Nrf2 or by suppressing the inflammatory gene expression by the inhibition of NFκ-B. These effects could be modulated by their bioactive compounds, such as polyphenols, flavonoids, tannins, saponins, and terpenes, present in these plants. This review summarizes the antioxidant and anti-inflammatory effects of traditional medicinal plants and highlights their molecular mechanisms of action and main bioactive compounds. This evidence may be used in biotechnology and synthetic biology areas for the development of new products from plant-derived compounds to reduce the high recurrence rates of UL.

## 1. Introduction

Urolithiasis (UL) is the presence of stones in the kidneys or urinary tract; this pathology shows high recurrence rates and complications [1,2]. Its pathophysiology involves physicochemical and biochemical processes in the urine, including pH values of 5–6.5 or >7.5, low urine volume (<1500 mL), and an imbalance between urinary molecules capable of forming crystals (promoters), such as high levels of oxalate and calcium, and molecules capable of blocking or reducing crystallization processes or crystal retention (inhibitors), such as low levels of citrate, magnesium, potassium, osteopontin (OPN), and Tamm-Horsfall protein [3,4,5]. These alterations generate urinary supersaturation and crystallization, which induces crystal precipitation. The crystals may be deposited in the calyces or pelvis of the kidney in free form or attached to the renal papillae [6]. The adhesion of the crystals promotes their internalization in renal epithelial cells, which causes crystallization, crystal retention, and stone nidus development [7]. The crystals that form in this process are mainly composed of oxalate (Ox) and/or calcium oxalate (CaOx) or calcium phosphate (CaPO_4_) [8]. The deposition of these compounds induces the production of reactive oxygen species (ROS), which in turn causes damage to different cellular biomolecules such as proteins, DNA, and lipids. The generation of this damage could be attributed to oxidative damage to renal epithelial lipids resulting from the high rate of urinary Ox and its deposition in the tubules and glomeruli and the increased gene expression and production of molecules related to oxidative stress (OS) and inflammation [9,10].

There are many medical and pharmacological treatments for this pathology [11,12,13,14]. However, the use of medicinal plants has emerged as an alternative for UL management due to their accessibility, low cost, and cultural traditions [15,16,17]. The World Health Organization (WHO) emphasizes enhancing knowledge about traditional medicine while strengthening evidence regarding the quality, safety, appropriate use, and efficacy of traditional and cultural practices [18,19]. Due to there not being much information about the molecular mechanisms of several medicinal plants, it is important to identify what has been shown to have antioxidant and anti-inflammatory properties, largely attributed to the presence of polyphenols, flavonoids, tannins, saponins, and terpenes, and compounds that can counteract the oxidative and inflammatory mechanisms that contribute to UL [20]. However, there are few studies that analyze the advances in the knowledge of plants with properties that can modulate the response in terms of the reduction in these alterations in cellular and animal models of UL [20,21,22,23,24,25,26,27,28].

Currently, there is a significant interest in integrating ancestral knowledge with scientific information, especially in the field of medicine. The traditional use of medicinal plants is based on culture, while biotechnology relies on evidence from scientific studies. Despite this, both approaches have the same purpose, using products with pharmacological effects for interventions against some pathologies [29]. This approach supports the compatibility of the scientific area with traditional medicine. However, there are still some barriers that biotechnology has not been able to understand due to the complex physiological environment in the pathological conditions. In this sense, the combined application of biotechnology and traditional medicine could expand the therapeutic characteristics of these applications [30]. Biotechnology can identify and improve the quality of the natural active principles present in medicinal plants so that they can be used more effectively in the design of studies to reduce adverse effects or enhance the therapeutic efficacy of traditional medicine without toxicity [31].

### 1.1. Oxidative Stress and Inflammation: Processes That Promote Urolithiasis

OS is defined as an imbalance between ROS and antioxidants [32,33]. In UL, renal epithelial cells are exposed to high CaOx concentrations, leading to pathological biomineralization and increased ROS production, which can trigger OS [32,34]. In response to OS, cells activate redox-sensitive transcription factors, such as nuclear factor 2 (Nrf2), which promotes the expression of endogenous antioxidant genes [32]. This antioxidant system is composed of different antioxidant enzymes, such as superoxide dismutase (SOD), catalase (CAT), and glutathione peroxidase (GPx), as well as the synthesis of the antioxidant compound glutathione (GSH) [33,35,36]. Under pathological conditions, such as UL, OS is exacerbated by a decrease in the endogenous antioxidant system [35]. Elevated ROS levels cause oxidative damage to biomolecules, including DNA, proteins, and lipids, leading to tissue dysfunction [37,38]. One of these processes is lipoperoxidation, which promotes the oxidation of polyunsaturated fatty acids and generates toxic molecules, such as 4-hydroxynonenal or malondialdehyde (MDA) [34,39]. The CaOx crystal deposition in the kidneys is associated with renal epithelial injury (changes in epithelial function and structure leading to crystal adhesion and increased ROS) and inflammation, which promote the expression of kidney injury molecule-1 (KIM-1) in models of hyperoxaluria. In patients with UL, KIM-1/creatinine has been correlated with stone size [40,41,42]. KIM-1 is a type I membrane glycoprotein present in renal tubules that contains immunoglobulin and mucin-like extracellular domains, with a transmembrane domain and a short intracellular domain. During pathological conditions, the secretion of inflammatory cytokines, which contributes to the recruitment of immune cells to the site of injury, is associated with macrophage aggregates and profibrotic areas with increased expression of α-smooth muscle actin, a marker of myofibroblast transformation [43,44]. ROS also act as signaling molecules that trigger stress responses to induce gene expression, activating biological mechanisms that protect cells and tissues against damage caused by various stressors. For instance, when phosphorylated, the kappa-light chain enhancer of activated B cells (NF-κB), which is translocated to the nucleus when phosphorylated, regulates the expression of pro-inflammatory genes, such as interleukin-1β (*Il-1β*), interleukin-6 (*Il-6*), and tumor necrosis factor-alpha (*Tnf-α*) [32,45,46,47,48]. Furthermore, increased inflammatory molecules and macrophage activation by ROS damage secreted cytokines, such as monocyte chemoattractant protein (MCP-1) and OPN [23,34,49]. This pro-inflammatory environment causes infrarenal alterations in cellular components, generating pathological changes in the kidney during UL, such as a deformed renal capsule and a narrow or absent capsular cavity, proximal tubules with swollen epithelial cells, numerous vacuoles in the cells, absence of brush border and necrotic cell fragments, and interstitial edema in the tubules [50,51,52] (Figure 1).

### 1.2. Traditional Medicinal Plants: Alternatives for Urolithiasis Management

For centuries, traditional medicine has been an integral resource for health in households and communities. The WHO defines traditional medicine as “the body of knowledge, skills, and practices based on indigenous theories, beliefs, and experiences of different cultures, whether explicable or not, that are used for the maintenance of health as well as for the prevention, diagnosis, amelioration, or treatment of physical and mental illness” [53].

According to the WHO, the use of and demand for traditional medicine are growing, particularly among people living in remote and rural areas where it is often the first choice for health and well-being due to its accessibility, affordability, and cultural acceptance [54]. Moreover, current general development trends in developing countries with increasing populations, poor coverage of Western medical care and access to traditional medicines and developed countries with aging populations indicate that the consumption of medicinal plants is not likely to decrease in the short to medium term [55].

Different ethnicities, cultures, and civilizations around the world have used medicinal plants as part of their traditional systems to treat various diseases and ailments such as gastrointestinal disorders, musculoskeletal disorders, colds, coughs and sore throats, injuries, respiratory system disorders, dermatological infections, and urinary system disorders, including UL [56,57,58,59,60,61,62]. Studies of medicinal plants for UL management have been documented from different countries, which have highlighted that ethnomedicinal traditions have therapeutic value and are important biological resources that have demonstrated urological benefits, such as stone expulsion, increased diuresis, and anti-spasmodic effects [22,23,63,64]. These benefits are associated with the presence of different bioactive compounds that have antioxidant effects in these traditional medicinal plants [25,65,66,67].

This information provides new perspectives for the use of traditional medicinal plants, which have gained increasing attention. A detailed study of their properties, bioactive compounds, and possible mechanisms of action presents opportunities for the development of evidence-based products that can be applied in the pharmacological and health area, especially for the UL population. Thus, the aim of the present review was to summarize studies showing the antioxidant and anti-inflammatory effects of traditional medicinal plants used in UL management through a review of the literature of the current knowledge.

## 2. Results and Discussion

The present review includes seven studies that evaluated the antioxidant and anti-inflammatory effects on UL. No preventive studies were included. These studies investigated six traditional plant species commonly used in UL treatment (Appendix A).

### 2.1. Antioxidant and Anti-Inflammatory Effects of Traditional Medicinal Plants in Urolithiasis

The plants described in this review are *Glechoma longituba* (*G. longituba*), *Bergenia ligulate* (*B. ligulate*), *Lygodium japonicum* (*L. japonicum*), *Citrus limon* (*C. limon*), *Xanthium strumarium* (*X. strumarium*) and *Tribulus terrestris* (*T. terrestris*) [68,69,70,71,72,73]. Among these, two plants (*G. longituba* and *B. ligulata*) were evaluated in vitro in a human kidney epithelial cell line (HK) [68,69]. The other plants were evaluated on animal models of UL using ethylene glycol (EG) alone or in combination with ammonium chloride (NH_4_Cl) for crystal generation (Table 1).

**Table 1 plants-14-02032-t001:** Antioxidant and inflammatory effects of traditional medicinal plants in urolithiasis models.

Scientific Name	Common Name	Region	Study Design	Antioxidant Biomarkers	Inflammatory Biomarkers	Reference
			**in vitro**			
*Glechoma longituba*	Naki	Beijing, China	Cell lines HK-2 were incubated with CaOx crystals (67 μg/cm^2^) and the effect of the Aq extract of the dried aerial part of *Glechoma longituba* (AExGl) was evaluated for 24 h. C (medium) CaOx (crystals: 67 μg/cm^2^) CaOx + Potassium citrate at different amounts: CaOx + AExGl (0.5 mg/mL) CaOx + AExGl (1 mg/mL) CaOx + AExGl (2 mg/mL) CaOx + AExGl (4 mg/mL)	**CaOx vs.** CaOx + 1 ↓ MDA ↑ SOD CaOx + 2 and CaOx + 4 ↓ MDA ↑ SOD, CAT **Potassium citrate vs.** CaOx + 1 ↓ MDA ↑ SOD CaOx + 2 and CaOx + 4 ↓ MDA ↑ SOD, CAT	**CaOx vs.** CaOx + 0.5, CaOx + 1, CaOx + 2 and CaOx + 4 ↓ KIM-1 ↓ OPN **Potassium citrate vs.** CaOx + 1, CaOx + 2, CaOx + 3 and CaOx + 4 ↓ KIM-1 ↓ OPN	[68]
*Bergenia ligulata*	*Bergenia ligulata* (*Saxifragaceae*)	Chandni Chowk, New Delhi, India	Cell lines HK-2 were incubated with Na_2_Ox (2 mM) and evaluated the effect of the EtOH extract of plant and rhizome *Bergenia ligulata* (EexBl) for 24 h. C (DMEM medium) Ox (2mM) Ox (2mM) + Cystone (200 μg/mL) Ox (2mM)+ EexBl (200 µg/mL)	**Ox vs.** EexBl + Ox ↓↓↓ H_2_O_2_	**Ox vs.** EexBl + Ox ↓↓↓ *Opn,* (*Mapk14-p38* MAPK), *Nfkb*	[69]
**In vivo**						
*Lygodium japonicum*	*Lygodium japonicum*	Ulsan, Corea	Wistar rats were induced to UL with EG (75%) ad libitum for 28 days. The animals were divided into the following experimental groups (*n* = 6 per group). Treatment with EtOH extract of *Lygodi Spora* (EexLS) was administered orally for 28 days. C (water) EG EG + distilled water (1 mL) EG + EexLS (400 mg/kg BW)	In kidney **EG vs.** EG + EexLS ↑↑ SOD ↓↓ MDA **EG + distilled water vs.** EG + EexLS ↑↑ SOD ↓↓ MDA	In kidney **EG vs.** EG + EexLS ↓↓ % Chronic inflammation	[70]
*Citrus limon*	Lemon peel	China	Wistar rats were induced to UL with EG (75%) ad libitum for 30 days. The animals were divided into the following two experimental groups (*n* = 6 per group). Treatment with aqueous-MeOH extract of lemon peel (MExLP) daily for 20 days. C (water) EG EG + MExLP (100 mg/kg BW)	In kidney **EG vs.** EG + MExLP ↓↓↓ MDA	In kidney **EG + 100 vs.** EG ↓↓↓ OPN ↓↓↓ *Nfkb*, *Tnf-α, Il-6*	[71]
*Xanthium strumarium*	Common Cocklebur, Donkeybur	India	Wistar rats were induced to UL with EG (75%) and NH_4_Cl (1%) ad libitum for 14 days and EG alone for the next 14 days. The animals were divided into the following experimental groups (*n* = 6 per group). Treatment with EtOH-Aq extract of the *Xanthium strumarium* fruit (AExXs) was administered orally from 14 days. C (water) EG + NH_4_Cl EG + NH_4_Cl + Water (vehicle control) EG + cystone (100 mg/kg BW) EG + AExXs (500 mg/kg BW)	In kidney **EG + NH_4_Cl vs.** EG + AExXs ↓↓ MDA ↑↑ CAT, SOD EG + Potassium citrate ↓↓ MDA ↑↑ CAT, SOD	In kidney **EG + NH_4_C**l **vs.** EG + 500 mg/kg ↓↓ % OPN EG + Potassium citrate ↓↓ %OPN	[72]
*Tribulus terrestris*	*Tribulus terrestris*	Bangalore, India	Wistar rats were induced to UL with EG (0.4%) with NH_4_Cl (1%) for 15 days and EG (4%) from the 16th to 28th day. The animals were divided into the following experimental groups (*n* = 6 per group). Treatment with Aq extract of fruit of *Tribulus terrestris* (AExTt) administered orally from the 16th to 28th day. C (water) EG + NH_4_Cl EG + NH_4_Cl + Vehicle EG + cystone (750 mg/kg BW) EG + NH_4_Cl+ AExTt (75 mg/kg BW) EG + NH_4_Cl+ AExTt (225 mg/kg BW) EG + NH_4_Cl+ AExTt (750 mg/kg BW)	In kidney **EG + NH_4_Cl vs.** EG + NH_4_Cl +75, EG + NH_4_Cl + 225, EG + NH_4_Cl 750 ↓↓↓ MDA, CAT ↑↑↑ GSH	In kidney **EG + NH_4_Cl vs.** EG + NH_4_Cl + 750 ↓↓↓ *Mapk14* and p38MAPK	[73]

Differences between groups are shown by *p* values: one arrow *p* < 0.05; two arrows *p* < 0.01; and three arrows *p* < 0.001. UL: urolithiasis; EG: ethylene glycol; NH_4_Cl: ammonium chloride; MeOH: methanolic extract; EtOH: ethanolic extract; and Aq: aqueous extract. HK-2: human kidney epithelial cell line; H_2_O_2_: hydrogen peroxide; Opn: osteopontin; GPx: glutathione peroxidase; MDA: malonaldehyde; GSH: glutathione; CAT: catalase; SOD: superoxide dismutase; *IL-6*: interleukin 6; MCP-1: monocyte chemoattractant protein-1; *TNF-α*: tumor necrosis factor alpha; IL-1-β: interleukin-1 beta; DMEM: Dulbecco’s Modified Eagle’s Medium; and BW: body weight.

#### 2.1.1. Antioxidant Effect of Traditional Medicinal Plants on Urolithiasis

Studies in vitro have shown the antioxidant effects of plants, such as *G*. *longituba* and *B*. *ligulata*. In renal cells (HK-2), incubation with *G*. *longituba* (2 and 4 mg/mL aqueous extract from aerial parts) increased the antioxidant enzymes CAT and SOD, while 1 mg/mL only increased SOD. In addition, all concentrations of aqueous extracts from *G*. *longituba* (1, 2, and 4 mg/mL) decreased the lipoperoxidation marker MDA [68] (Table 1). *B. ligulata* (200 µg/mL ethanol extract from the plant and rhizome) showed a decrease in hydrogen peroxide (H_2_O_2_) in HK-2 [69]. Furthermore, preclinical studies have evaluated the effects of plants under various pathological conditions, such as UL, and different antioxidant biomarkers, such as SOD, MDA, CAT, and GSH. In animal models of UL, treatment with plants, such as *L*. *japonicum* (400 mg/kg BW ethanol extract from pores) [70], *C*. *limon* (100 mg/kg BW aqueous methanol extract from lemon peel) [71], *X*. *strumarium* (500 mg/kg BW hydroethanolic extract from fruit) [72] and *T*. *terrestris* (75, 225, and 750 mg/kg BW aqueous extract from dry and ripe fruit) [73] reduced the MDA levels in kidneys. This oxidative biomarker may be due to OS generated during UL, as it involves a decrease in the endogenous antioxidant system (Table 1).

After oral treatment with *L*. *japonicum* (400 mg/kg BW ethanol extract from spores) [70] and *X*. *strumarium* (500 mg/kg BW hydroethanolic extract from fruit) [72] there was an increase in the SOD enzyme in the kidneys. The use of *X*. *strumarium* (500 mg/kg BW hydroethanolic extract from fruit) increased the CAT enzyme [72]. Finally, *T*. *terrestris* (75, 225, and 750 mg/kg BW aqueous extract from dry and ripe fruit) increased the GSH levels after intervention [73].

#### 2.1.2. Anti-Inflammatory Activity of Traditional Medicinal Plants in Urolithiasis

Inflammation is a key process associated with alterations in UL. Thus, controlling inflammation plays an important role in the development and management of this pathology. The included studies showed that *G*. *longituba*, *C*. *limon*, *X*. *strumarium*, *T*. *terrestris*, and *L*. *japonicum* had an anti-inflammatory effect (Table 1).

The anti-inflammatory activity of traditional plants was assessed with different biomarkers, such as KIM-1, OPN, P38 MAPK, IL 6, MCP-1, TNF-α and Il-1β proteins, and *Nfkb*, *Tnf-α*, *Il-6*, and *Mapk14* genes. In vitro studies have shown a reduction in OPN and KIM-1 proteins with *G*. *longituba* (0.5, 1, 2, and 4 mg/mL aqueous extract from the aerial part) [68]. Another study in cells showed that *B*. *ligulata* (200 µg/mL ethanol extract from the plant and rhizome) reduced the expression of genes related to inflammation, such as *Opn*, *Mapk14*, and *Nfkb* [69].

Studies in animal models have shown that *C*. *limon* (100 mg/kg BW aqueous methanol extract from lemon peel) [71] and *X*. *strumarium* (500 mg/kg BW hydroethanolic extract from fruit) [72] decreased the inflammatory effects through OPN protein immunity. *C. limon* (100 mg/kg BW aqueous methanol extract) decreased the gene expression and protein level of NF-kβ, TNF-α, and IL-6 [71], *T. terrestris* (750 mg/kg BW aqueous extract of dry and ripe fruit) showed a decrease in *Mapk14* and p38MAPK protein [73]. Furthermore, *C*. *limon* (100 mg/kg BW aqueous methanol extract) decreased the percentage of inflammation in renal tissue [70].

## 3. Molecular Effects of Bioactive Compounds in Traditional Medicinal Plants on Urolithiasis

The studies included used plant extracts as treatment for UL. These extracts contained different types of bioactive compounds depending on the solvent used [74,75,76]. The present review briefly describes specific chemical characteristics of these bioactive compounds. The plant extracts in the included studies were obtained mainly with water and ethanol. Water extraction, mainly hydrophilic and polar compounds, are obtained from the plant material, including polyphenols, flavonoids, tannins, essential oils (terpenoids), saponins, organic acids, and some alkaloids, while steroid was mainly extracted with methanol [77,78,79,80].

### Antioxidant and Anti-Inflammatory Effects of Bioactive Compounds from Plant Extracts

In fact, the extracts of medicinal plants included in this review might contain these types of bioactive compounds that can have antioxidant and/or anti-inflammatory effects (Table 2).

Polyphenols, in particular flavonoids and tannins, have demonstrated the potential to reduce crystal formation through their antioxidant activity, directly through neutralized ROS [81]. Phenolic compounds can act as chelators of metals such as Fe^3+^, inhibiting the production of hydroxyl radicals (^•^OH) in the Fenton reaction [82,83]. Tannins inhibit pro-oxidant enzymes such as NADPH oxidase, which is considered the main source of OS and pro-inflammatory molecules [84]. These activities can interrupt the propagation stage of lipid auto-oxidation chain reactions, reduce lipid peroxidation and decrease oxidative damage and inflammatory mediators present in the UL [83,85,86,87]. On the other hand, compounds such as terpenoids, saponins, alkaloids, and sterols have also demonstrated endogenous antioxidant activity with the donation of methyl groups and direct capture of ROS or by the upregulation of antioxidative enzymes such as SOD, CAT, GR, GSH, and GPx [88,89,90].

The anti-inflammatory activity of these bioactive compounds present in medicinal plants is through the activation or silencing of genes encoding defensive enzymes, transcription factors, and structural proteins [77]. Polyphenols and flavonoids exert their effects mainly through the inhibition of the transcription factor NF-κB and the enzyme cyclooxygenase (COX) [91,92]. Tannins can inhibit phospholipase A2, Keap1, and NF-κB, interfering with the inflammatory cascade [85,86]. Terpenoids act by inhibiting macrophage proliferation and reducing the release of inflammatory mediators [93,94]. Saponins are associated with a decrease in TNF-α and IL-6 levels, as well as the inhibition of COX-2 and prostaglandin E2 [95]. In the case of alkaloids, a reduction in NF-κB and TNF-α levels has been observed, as well as the inhibition of β-glucuronidase secretion induced by a platelet-activating factor [88]. Steroids exert their anti-inflammatory effects through the inhibition of NF-κB and a reduction in nitric oxide (NO^•^) release and COX-2 activity [96].

**Table 2 plants-14-02032-t002:** Bioactive compounds in traditional medicinal plants used on urolithiasis management.

Scientific Name	Common Name	Plant Part	Qualitative Compounds	Compounds of Interest or Major Proportion	Method of Identification	Reference
*Glechoma longituba*	Nakai	Aerial part	Terpenoids Steroids Flavonoids Polyphenols Alkaloids	(2E)-3-3,4-Dihydroxyphenyl)-2-propenoyl|oxy} malonic acid	HPLC-HR MS	[97,98]
				Trans-caffeic acid *		
				Rosmarinic acid *		
				Luteolin-7-O-di-glucuronide		
				Apigenin-7-O-di-glucuronide		
				Luteolin-7-O-glucuronide		
				Apigenin-7-O-glucuronide *		
*Bergenia ligulata*	Paashanbheda, bheda, Ayurveda	Plant and rhizome	Alkaloids Glycosides Saponins Carbohydrates Phenols Flavonoids Diterpenes	Phenol, 2,4-bis(1,1-dimethylethyl)	GC-MS LC-MS	[99,100,101]
				Squalene *		
				Bergenin *		
*Lygodium japonicum.*	*Lygodium japonicum*		Phenols Glucosides Flavonoids	Methyl Protocatechuate Caffeic acid * Chlorogenic acid * Linarin Apigenin * Narimgerina Kaempferol * Quercetin *	Miniature mass spectrometry	[102]
*Citrus limon*	*Lemon*	Lemon peel	Phenols Flavonoids	Caffeic Ferulic * Hesperidin * Eriocitrin * Diosmin * Rutin * Cynarosides	HPLC UPLC-PDA	[103,104,105]
*Xanthium strumarium*	Common Cocklebur, Donkeybur	Fruit	Alkaloids Flavonoids Triterpenoids Terpenoids Tannins Saponins Quinones Coumarins Carbohydrates Glysides Phenolics	Chlorogenic acid *	HPLC GC–MS	[106,107,108]
				3-O-caffeoylquinic acid		
				1-O-caffeoylquinic acid		
				4-O-caffeoylquinic acid		
				1,3-O-dicaffeoylquinic acid		
				1,4-O-dicaffeoylquinic acid		
				1,5-O-dicaffeoylquinic acid		
				4,5-O-dicaffeoylquinic acid		
				1,3,5-O-tricaffeoylquinic acid		
				3,4,5-O-tricaffeoylquinic acid		
*Tribulus terrestris*	Gokharu	Fruit	Carbohydrates Amino acids and peptides Glycosides Tannins Terpenoids Phenols Saponins Alkaloids Flavonoids	Terrestrosin 1	UHPLC/Q-TOF MSE	[109,110,111]
				Polianthoside D		
				Parvispinoside B		
				Purpureagitosid		
				Desglucolanatigonin Il		
				F-gitonin		
				25R-tribulosin		
				Ginsenoside Rb		
				Tigogenin-3-O-b-D-xylopyranosyl-(1 fi 2)-[bD-xylopyranosyl-(1 fi 4)]-[a-L-rhamnopyranosyl-(1 fi 2)]-b-D-galactopyranoside		
				Phytol *		

***** Compounds reporting antioxidant or inflammatory effects [112,113,114,115,116,117,118,119,120,121,122,123,124,125,126,127,128,129].

## 4. Challenges and Future Directions

Global trends in medicinal plants research show that since 2001 there has been an increase in studies related to traditional medicinal plants, with a focus in the areas of pharmacology, toxicology, and medicine, and China and India being the main countries with more than 10,000 publications [26,130]. The increased study of medicinal plants is especially important, as the WHO recognizes the practice of traditional medicine and has highlighted the need to generate information to support its efficacy and safety through scientific studies to provide effective and safe treatments based on scientific evidence [131].

For this reason, rigorous ethnopharmacological studies are encouraged to understand the uses, dosages, sources, and preparation methods of traditional medicinal plants [132]. The consideration of this information will allow future studies to have a better knowledge of the plants used by communities. Therefore, more rigorous preclinical studies would allow greater therapeutic application [132,133]. Despite the extensive literature on medicinal plants and UL, clinical studies are limited [134,135]. It is important to note that future research in humans should be conducted through controlled studies, including the evaluation of adverse effects, and should consider different biomarkers to corroborate the effects in the study populations [20,136]. The literature highlighted that the causes of toxicity in herbal medicines are associated with incorrect information about identification, labeling, or standardization, as well as the presence of contamination with fungal toxins, such as aflatoxin [137]. In fact, preclinical evidence suggested that safety and toxicity were found in approximately 27% of 6000 herbal products marketed in 37 countries [138,139]. However, the guidelines have not always suggested the evaluation of acute and chronic toxicity in products of plant origin by remarkable heterogeneity in studies, particularly in terms of the part of the plant used and the extraction method employed [136,140]. Thus, optimizing the doses and formulations of medicinal plants will facilitate an adequate evaluation of their safety and guarantee a standardization in quality, providing greater scientific support [27,141].

Biotechnology and synthetic biology, based on scientific evidence of preclinical and controlled clinical studies, are fundamental in the development of new products from plant-derived compounds to reduce the high recurrence rates of UL [132,133,142]. In this sense, advances in bioproduction, genetic engineering, and route optimization are expected to expand the range of substances that can be extracted from plants, but this will require multidisciplinary, national, and international collaborations [143]. One of the areas related to the improvement of products of plant-derived traditional medicine is nanotechnology, through which processes have been applied for biomolecule production, which can improve their solubility, stability and therapeutic efficacy, as well as reduce their toxicity [144]. This strategy could be beneficial in plants used for UL, as it would optimize the release of bioactive compounds and improve their bioavailability and enhance their antioxidant, anti-inflammatory, and anti-litholytic effects; however, further research is still required.

## 5. Search and Inclusion Methods

### Search Strategy

The present scoping review was conducted following the Preferred Reporting Items for Systematic reviews and Meta-Analyses extension for Scoping Reviews (PRISMA-ScR) [145]. The protocol was registered with Open Science Framework (OSF) and is publicly available at https://osf.io/ky7e9, accessed on 5 June 2025, Two authors (B.P.-H. and A.A.-N.) performed the search strategy. The database sources used were MEDLINE/PubMed, Google Scholar, SpringerLink, Scielo, and Redalyc, with studies published from 2000 to September 2024. The search strategies used were the keywords: “medicinal plants”, “herbal medicine”, “plants”, “traditional medicine”, “plant extracts”, AND “antioxidant”, “oxidative stress” “anti-inflammatory”, “inflammation”, “oxidative damage”, AND “kidney stones”, “urolithiasis”, “calcium oxalate”, “urinary calculus”, NOT “review”. We applied the Population, Intervention, Control, Outcome (PICO) strategy (Table 3). Here, lithogenic biomarkers refers to crystal-induced injury and urinary parameters (volume, pH, citrate, calcium, uric acid, and magnesium); the antioxidant biomarkers refers to antioxidant enzymes, such as superoxide dismutase (SOD), catalase (CAT), and molecules such glutathione (GSH) and malondialdehyde (MDA); and inflammatory biomarkers such as kidney injury molecule-1 (KIM-1), osteopontin (OPN), monocyte chemoattractant protein-1(MCP-1), nuclear factor kappa-light chain enhancer of activated B cells (NF-κB), tumor necrosis factor-alpha (TNF-α), and interleukin-6 (IL-6).

The eligibility criteria for the articles were scientific articles that complied with (1) studies in models of UL in vivo and in vitro; (2) the evaluation of an extract of plant (roots, leaves, seeds, bark, or other constituent parts); (3) the use of plant extract as treatment management; (4) the report of lithogenic, antioxidant, and inflammatory biomarkers; and (5) were in the English or Spanish language. Exclusion criteria were: (1) the evaluation of polyherbal extracts; (2) the use of plant extract as preventive management; and (3) review articles, books, theses, or conference proceedings.

Data extraction was carried out using a structured recording form developed specifically for this review, based on the methodological guidance provided by the Joanna Briggs Institute (JBI) for scoping reviews [146]. Data extraction from all the selected articles was carried out independently. The following elements were considered: 1. Scientific name of plants, 2. common name of plant, 3. origin/country of origin, 4. type of intervention, 5. comparator, 6. population and sample size, 7. duration of intervention, 8. main outcomes related to the scoping review questions, and 9. author(s) and year of publication. All included studies were independently reviewed by two authors, and data were recorded accordingly. In the search, 365 studies were identified, of which 181 were excluded as duplicates, books, or reviews. An additional 54 studies were removed after title screening, leaving 184 studies for potential inclusion. After abstract screening, 124 studies were excluded based on predefined criteria, primarily due to toxicity studies, evaluations of metabolites or polyherbal extracts, in vitro chemical assays, preventive approaches, and studies that evaluated the lithogenic process, and anti-inflammatory and antioxidant effects. Finally, six studies were included in the review (Figure 2).

## 6. Conclusions and Recommendations

This scoping review provides us with relevant information from preclinical studies of plants proven to have therapeutic effects on the lithogenic process, and antioxidant and inflammatory biomarkers published between 2000 and 2024. The information summarized from these studies focuses on the antioxidant and anti-inflammatory effects of traditional medicinal plants used in UL management, highlighting their main bioactive compounds and potential molecular mechanisms. Evidence shows that medicinal plants have different active compounds, such as polyphenols, flavonoids, tannins, saponins, terpenes, and steroids, which induce beneficial therapeutic effects directly or indirectly, resulting in their use as medicinal agents in the treatment of UL. The synergy of their bioactive compounds makes them promising alternatives to traditional pharmaceuticals, considering that their toxic doses evaluated range from 1500 to 5000 mg/kg BW without toxicity effects, except for *G. longituba*, which has no toxicity studies. However, it is crucial to address the challenges related to the standardization, dosage, quality, and safety of these medicinal plants to fully realize their potential as valuable sources of nutraceuticals. Therefore, it is recommended that future research incorporates a more comprehensive evaluation of biomarkers to enhance the scientific foundation and support the translational advancement of phytotherapeutic strategies for the management of UL.

## 7. Limitations

Among the limitations of the present review is the small number of studies evaluating antioxidant and inflammatory biomarkers, even though these mechanisms are key in the pathophysiology of UL. This lack of information impacts the understanding of the molecular mechanisms involved, which could hinder future clinical application. Another limitation is the heterogeneity of the medicinal plant parts used, as well as the methods used to obtain the extracts that were tested as an intervention against UL. Also, some included studies did not provide detailed information on the bioactive compounds present in the extracts. In addition, the risk of bias and the quality of the included studies were not assessed.

## Figures and Tables

**Figure 1 plants-14-02032-f001:**
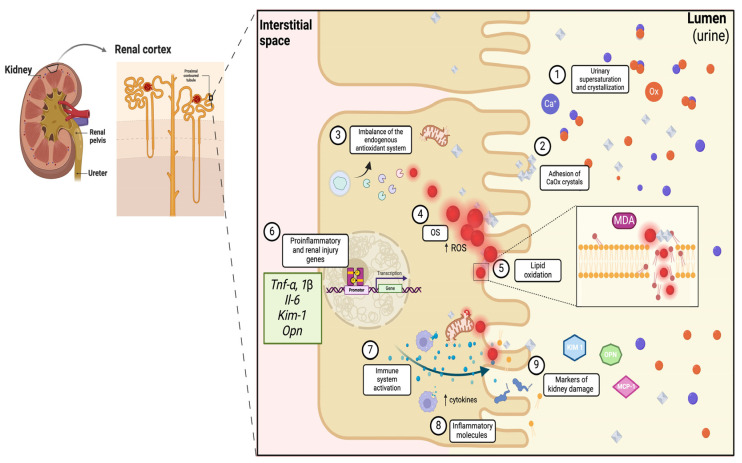
Molecular mechanism of oxidative stress and inflammation on urolithiasis. 1. Oversaturation and crystallization of calcium and oxalate crystals in urine; 2. growth and adhesion of CaOx crystals to epithelial cells of the renal proximal growth and adhesion of CaOx crystals to renal proximal tubular epithelial cells; 3. infiltration of the crystals promotes the imbalance of the endogenous antioxidant system; 4. generation of endogenous antioxidant system causes an increase in ROS leading to EO; 5. promotion of lipid oxidation; 6. synthesis of pro-inflammatory and renal injury genes; 7. activation of the immune system; 8. increase in pro-inflammatory cytokines; and 9. increase in markers in urine. CaOx: calcium oxalate; OS: oxidative stress; ROS: reactive oxygen species; MDA: malondialdehyde; KIM-1: kidney injury molecule-1; OPN: osteopontin; and MCP-1: monocyte chemoattractant protein-1. Created with BioRender.com.

**Figure 2 plants-14-02032-f002:**
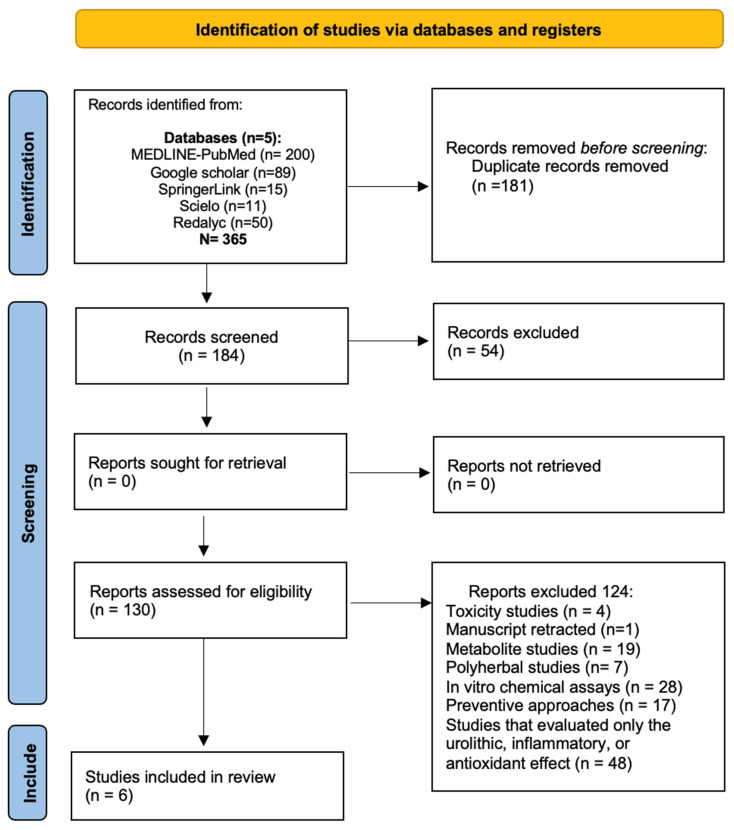
Flow chart (PRISMA) of studies included.

**Table 3 plants-14-02032-t003:** PICO criteria for study selection.

	Criterion	Description
P	Population	Models of UL
I	Intervention	Use of traditional medicinal plant (roots, leaves, seeds, bark, or other constituent parts)
C	Comparator	Any comparator positive or negative or control group
O	Outcomes	Lithogenic, antioxidant, and inflammatory biomarkers

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
