# Peer review of "Antioxidant and Anti-Inflammatory Effects of Traditional Medicinal Plants for Urolithiasis: A Scoping Review"

_plants, 2025, doi:10.3390/plants14132032_

Round 1
Reviewer 1 Report (Previous Reviewer 2)
Comments and Suggestions for Authors
The authors have corrected the manuscript and is suitable for publication in its current form
Reviewer 2 Report (Previous Reviewer 3)
Comments and Suggestions for Authors
The modified version is suitable for publication in this journal and is recommended to be accepted in its current form
Reviewer 3 Report (Previous Reviewer 4)
Comments and Suggestions for Authors
Dear Authors,
After a thorough reworking of the article, I can recommend it for publication in the journal Plants.
This manuscript is a resubmission of an earlier submission. The following is a list of the peer review reports and author responses from that submission.
Round 1
Reviewer 1 Report
Comments and Suggestions for Authors
The narrative review expands on the knowledge of traditional medicinal plants and their anti-inflammatory and antioxidant effects for the treatment of the urinary system disease, Urolithiasis. Very few literature studies have investigated the prospects of medicinal plants for the treatment of urolithiasis, highlighting a promising scope of the literature review in the present context of natural product-mediated drug discovery and development.
However, substantial changes are required for a clear and concise representation, prioritized actionable goals, discussion of emerging biotechnologies in product development, and future directions in this area.
Abstract: The abstract provides an overview of the traditionally used medicinal plants and their phytoconstituents. What is the recent progress or biotechnological advances in this direction? In addition, current progress/achievements and future directions in the use of medicinal plants for Urolithiasis need to be discussed briefly.
The medicinal plants Glechoma longituba, Bergenia ligulata, Lygodium japonicum (Thunb.) Sw., Citrus lemon, Xanthium strumarium, Tribulus terrestris and Pyrrosia petiolosa and the bioactive metabolites were investigated for their anti-urolithiasis effects. What is the rationale for focusing on this kidney disease? Explain.
How does the present study contribute to improving our knowledge of plant-based therapeutic alternatives?
Introduction section: The plant names and brief description of the phytocontituents are required. The intro should be elaborated to include the pharmaceutical prospects of these medicinal plant species, recent case studies, limitations in product development, etc.
Material and method section: Research Methodology is not discussed. Add a flowchart or table summarizing the PRISMA-style selection process, including the number of papers screened, excluded, and analyzed for the review preparation.
The authors have discussed the potent efficacies of 7 traditional medicinal plants against Urolithiasis. Are there any side effects observed in the plant extract intake? Considering the potential side effects of the medicinal plants. Discuss.
Figures and Tables: All the figures/diagrammatic representations are well presented and discuss key pieces of information.
Conclusion: The concluding remark must include the key outcomes of the study. Please discuss the translational achievements in plant product development (if any), major challenges, and future prospects of the plants under study.
Line 360-362: However, it is crucial to address………… standardization, dosage, quality and safety….Any key studies in this direction? Please discuss briefly.
Minor suggestions:
Line 37-39: The sentence should be revised for clarity. The sentence should not be too long and should clearly express the underlying meaning.
Figure 1. Discuss the molecular mechanisms of oxidative stress and inflammation in urolithiasis. Please improve the resolution of the image to at least 300 dpi., The image seems blurred.
Moderate English revisions are required to improve the readability of the literature.
Comments on the Quality of English LanguageModerate English revisions are required to improve the readability of the literature.
Reviewer 2 Report
Comments and Suggestions for Authors
Please find the attached.
Thank you

Reviewer 3 Report
Comments and Suggestions for Authors
1. The review section on "Traditional Medicinal Plants Used for Treating Urinary Stones" should be expanded to summarize which traditional medicinal plants are recommended for UL treatment.
2. Why only choose the seven medicinal plants for review? What is the connection between them? What are the important aspects of these plants worth paying attention to? Clear topic selection criteria should be provided. The lack of reasons and medicinal background introduction of these plants here is currently confusing.
3. The comprehensive review of seven medicinal plants used in the treatment of UL is inadequate both in scope and depth.
In general, it is deemed that there is considerable potential for enhancing this manuscript, and given its current standard, it is not advisable to publish it in this journal.
Reviewer 4 Report
Comments and Suggestions for Authors
Lines 22-23: "Thus, the aim of the present review was to summarize the studies that showed antioxidant and anti-inflammatory effects of traditional medicinal 23 plants used in UL management". This is a confusing sentence. Can all antioxidant and anti-inflammatory plants be against UL? This needs correction. Many plants possess that kind of activity.
Line 27: other Latin names are without authors ((Thunb.) Sw.). Please standardize the style.
Chapters 1-3: Sufficiently concise and relevant.
Line 145: Really only in México, China and India? Please clarify.
Figure 2: This is a good illustration, but it seems to be more a graphical abstract than a part of the review.
Chapters 5.1 and 5.2: This is relevant, but do only the mentioned plants demonstrate antioxidant and anti-inflammatory activities? This is quite common in the plant kingdom.
Lines 223-225: "The effects of traditional medicinal plants are associated with the presence of bioactive compounds [81]. These bioactive compounds can activate mechanisms by themselves or secondary metabolites [82]." Such elementary truths cannot be part of a scientific article.
Lines 229-230: "According to the literature reviewed, only information on bioactive compounds is presented in five of the seven plants included." Are the authors really sure that nothing is known about the chemical composition of the other two plant species?
Page 11: too much empty area on the page. Please modify the layout.
Table 2: Why is there nothing about Citrus lemon chemical composition?
6.1 Polyphenols: "Polyphenols or some main classes are reported in five of the plants describe in this review (G. longituba, B. ligulata, X. strumarium L., T. terrestris and P. petiolosa) (Table 2)." Why only in five of the plants? Polyphenols exist in each higher plant species.
Lines: 238-245. This information is too general and belongs to the ABC of medicinal plants.
How is UL related to polyphenols?
6.2 Flavonoids: The same question about the contents of other plants here. Also, too general and well-known information again.
Line 264: "coumarin" is not a flavonoid.
6.3-6.7: The same problems.
Conclusions: Too general. What do we know new now? How to continue their study?
Which other plants have UL activity?
References: Only 115 references are not enough for a review article.
My main concern: Such elementary truths cannot be part of a scientific article. The treatment is too general and does not give us remarkably new knowledge. The concept of the article raises doubts. Hundreds and hundreds of review articles of this superficial nature could be written.
